materials science/nanotechnology

Tween 80, nanoemulsion, reduction reaction, silver nanoparticle

**Author for correspondence:**
Toshiharu Enomae
e-mail: enomae.toshiharu.fw@u.tsukuba.ac.jp

This article has been edited by the Royal Society of Chemistry, including the commissioning, peer review process and editorial aspects up to the point of acceptance.

# Characterization of self-assembled silver nanoparticle ink based on nanoemulsion method

Donghao Hu[1], Kazuyoshi Ogawa[2], Mikio Kajiyama[2] and Toshiharu Enomae[2]

[1]Graduate School of Life and Environmental Sciences, and [2]Faculty of Life and Environmental Sciences, University of Tsukuba, 1-1-1 Tennodai, Tsukuba, Ibaraki 305-8572, Japan

TE, 0000-0001-9041-7528

A well-dispersed self-assembled silver nanoparticles (AgNPs) ink with high purity was synthesized via $AgNO_3$ emulsion prepared by blending an $AgNO_3$ aqueous solution and a liquid paraffin solution of both polyoxyethylene (20) sorbitan monooleate (Tween 80) and sorbitan monooleate (Span 80). The ink remained as an emulsion at low temperatures; however, it produced AgNPs after sintering at about 60°C and showed a high stability at nanoscale sizes (with diameters ranging 8.6–13.4 nm) and a high conductivity. During the whole procedure, Tween 80 acted as a surfactant, reductant and stabilizer. Presumably, Tween 80 underwent an autoxidation process, where a free radical of an α-carbon of ether oxygen was formed by hydrogen abstraction. The mean diameter of emulsion droplets could be reduced by decreasing water content and increasing the ratio of surfactant and concentration of $AgNO_3$ aqueous solution. Consequently, the thermogravimetric analysis and X-ray diffraction result clarified the purity of the produced $Ag^0$. Dynamic light scattering and ultraviolet-visible spectroscopy clarified that an increased concentration of $AgNO_3$ decreased the particle size.

## 1. Introduction

Electroconductive ink has remarkable significance in printing or drawing circuits for flexible electronics [1,2]. Carbon black [3] or carbon nanotube-epoxy resin composites [4,5] have been employed as representative electroconductive inks for screen printing in the early times of printed electronics. Electroconductive polymers, such as poly(3,4-ethylenedioxythiophene)–poly(styrenesulfonate) (PEDOT-PSS) [6] have also been used to fabricate transparent electrodes. Examples of silver-based inks include silver

**Figure 1.** Schematic of self-assembled AgNPs obtained via the nanoemulsion method.

nanoparticles dispersed in higher alkane liquids [7], silver nanowires [8] with better contacts among each other providing high conductivity, silver nitrate aqueous solutions with reducing agents [9] and silver species composites [10].

Unlike large silver particles, silver nanoparticles (AgNPs) with a mean diameter less than or equal to 100 nm have attracted continuous research attention owing to their unique physical and chemical properties, for instance, their high electrical and thermal conductivities, outstanding optical qualities and antibacterial properties [11–14].

Nowadays, there are three typical methods of fabricating AgNPs for ink-jet printing [11,15]: (i) physical approaches, (ii) chemical approaches, and (iii) green synthesis. The primary techniques using polymers in chemical approaches are [16–18] based on (i) mechanical mixing of a polymer and metal particles, (ii) polymerization with metal particles, and (iii) chemical reaction of metal salts.

Nanoemulsions are a new class of emulsions, with very small droplet dimensions on the nanometric scale [19–22]. Typically, nanoemulsions are transparent or translucent dispersions due to their characteristic droplet size (diameter less than 200 nm) [19,21].

Several studies present ideas that water in oil (W/O) emulsions or nanoemulsions could provide a microreactor system [23–26], in which reactions could occur inside water droplets, protected by an oil phase on a nanoscale.

Generally, the synthesis of nanomaterials by the conventional emulsion method requires two kinds of solutions or emulsions: oxidant and reductant types [27–29]. However, this multiple-step operation would incur an unnecessary wastage of chemicals and time.

Self-assembly is defined as the spontaneous association of components into structures without human intervention [30]. From a thermodynamical point of view, this novel method can be described as an equilibrium process, driven by the minimization of the Gibbs free energy [31]. One of the self-assembling methods involves the use of a droplet of colloidal suspension as a template [31–33], which provides immense potential for the future development of one-step synthesized AgNP inks.

In this study, a self-assembled AgNP nanoemulsion was synthesized via the nanoemulsion method, using polyoxyethylene sorbitan monooleate (Tween 80), which functions as a surfactant, reductant and stabilizer as shown in figure 1, and was used as a stable, highly conductive, self-assembled AgNP ink. The multifunctionality of Tween 80 could simplify and control AgNP generation, wherein a silver nitrate solution is the sole reagent needed in addition to the surfactants.

## 2. Materials

Tween 80, liquid paraffin (JIS Special Grade), ethanol (JIS Special Grade, 99.5%) and silver nitrate (JIS Special Grade) were all purchased from Wako Pure Chemical Industries, Ltd. Sorbitan monooleate (Span 80) purchased from Tokyo Chemical Industry Co. Ltd was used for the preparation of inks.

All chemicals were employed without any further purification, and deionized water was used for all experiments.

# 3. Experimental

## 3.1. Preparation and characterization of nanoemulsion

The stability of the nanoemulsion was tested by titrating liquid paraffin solutions of the surfactant blend, as a component with water. This served as a model of the $AgNO_3$ aqueous solution for determining the maximum emulsifiable limit, which can help in identifying the appropriate dose of $AgNO_3$ solution, both as a component and a medium. Tween 80 and Span 80 were dissolved in paraffin together, in different ratios (0:100, 20:80, 30:70, 40:60, 50:50, 60:40, 70:30, 80:20 and 90:10), but with a constant surfactant blend-paraffin ratio of 5:95 (w/w).

Emulsions were formed by adding water to the liquid paraffin solutions of the surfactant blend and stirring with a mixer (Automatic Labo-Mixer NS-8, PasolinaNS-8, Pasolina, Japan) for homogenization at 800 r.p.m. at room temperature for 30 min. The maximum water solubility was determined by observing the emergence of phase separation.

The droplet size distribution of the W/O emulsion was evaluated by dynamic light scattering (DLS) at 25°C, using a DLS spectrophotometer (DLS-7000, Otsuka Electronics Co., Inc., Osaka, Japan). The data were obtained as a mean value from four or five measurements and conducted immediately after the homogenization.

## 3.2. Synthesis and purification of silver nanoparticles

First, 2 ml aqueous $AgNO_3$ solutions with different concentrations (0.10, 0.20, 0.30 and 0.50 g ml$^{-1}$) were added into liquid paraffin solutions of the surfactant blend containing 6 g paraffin, 1 g Span 80 and 1 g Tween 80, and mixed with a magnetic stirrer at 500 r.p.m. at room temperature for a few minutes to form an $Ag^+$ nanoemulsion. Brackets [] will be used to indicate the concentration of the $AgNO_3$ solution in g ml$^{-1}$ in the text and figures hereafter.

For the synthesis of AgNPs, the AgNPs nanoemulsion was heated at 80°C and stirred at 500 r.p.m. for 15 min with a hot stirrer.

Purified solid AgNPs were obtained by a subsequent purification procedure with ethanol three times to remove surfactants and paraffin. Ethanol was added to each aqueous suspension of unpurified AgNPs; the sample was shaken thoroughly with a mixer and then centrifuged at 9060g RCF (relative centrifugal force) at 20°C for 20 min. Sedimentary solid particles were recovered by decantation, and further purification was carried out by repeated centrifugation. The sample was dried under $N_2$ gas for the later characterization.

## 3.3. Characterization of silver nanoparticles

The dried solid AgNP samples were analysed by X-ray diffraction (XRD, AXS D8 Advance, Bruker, Germany) in order to identify the crystal structure in a $2\theta$ range from 10 to 90° at a $2\theta$ scan speed of 0.2° s$^{-1}$ at 40 kV and 40 mA using Cu-K$\alpha$ radiation (1.5418 Å).

The degradation behaviour of the AgNPs was investigated by thermogravimetric analysis (TGA, Mettler Toledo TGA/SDTA 851, Gießen, Germany). A degradation diagram was obtained by placing 6.000 mg purified AgNPs into a platinum pan and heating from 40°C to 1100°C at the rate of 20°C min$^{-1}$. The Ar gas flow rate was 200 ml min$^{-1}$.

Fourier transform infrared (FTIR) spectroscopy was conducted to characterize the AgNPs with a FTIR spectrophotometer (FT/IR-6100, JASCO Corporation, Tokyo, Japan) at wavenumbers ranging from 600 to 3700 cm$^{-1}$ at room temperature. An unpurified liquid sample of Tween 80 was subjected to measurement by pasting a trace amount on the surface of a KBr single pellet. Solid AgNP samples were kept in a desiccator for drying for 12 h, followed by measurement with a pellet containing AgNPs and KBr (1:100).

Transmission electron microscopy (TEM, H-7650, Hitachi, Japan) was conducted to determine the static particle diameter ($d_S$) distribution of AgNPs at an acceleration voltage of 200 kV. Samples were prepared by placing a drop of the purified AgNP ethanol dispersion on a carbon-coated formvar Cu

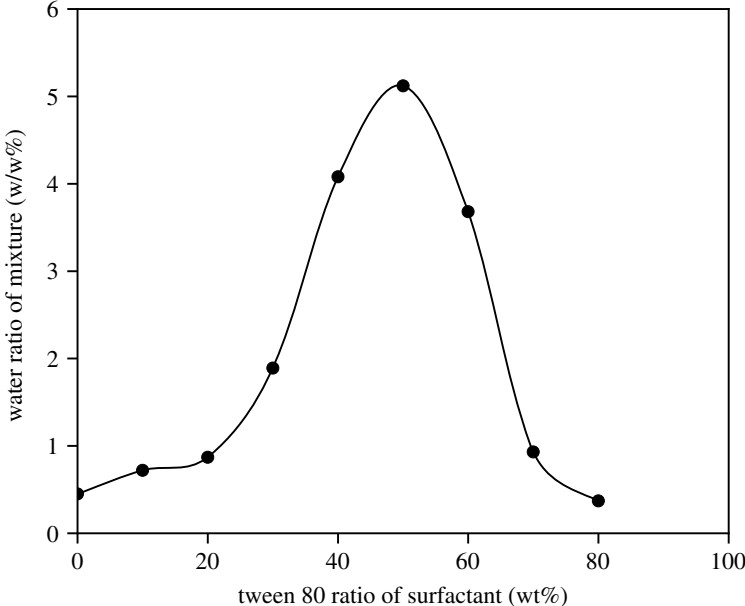

**Figure 2.** Maximum emulsifiable water content at different ratios of Tween 80 to Span 80.

grid and drying under nitrogen gas flow for approximately 10 min at room temperature. ImageJ was used for particle diameter analysis in a projected area range from 10 to 500 nm$^2$.

The ultraviolet (UV)-visible spectrum of the unpurified AgNP aqueous solution was acquired using a spectrophotometer (UV-3100 PC, Shimadzu, Kyoto, Japan) with a 3 ml solution in a quartz cell, with a light path length of 10 mm, at a wavelength interval of 200–800 nm, at room temperature.

The zeta potential was measured to demonstrate the stability of the AgNPs, with a zeta potential analyser (Zetasizer Nano-ZS, Malvern Instruments Ltd, UK) at 20°C, after sonication for 20 min at room temperature. The produced AgNPs were diluted 20 times. A 1.0 ml of the diluted AgNPs was shaken well, poured into a cuvette with a capacity of 1.0 ml and set to a sample chamber of the instrument (DTS1070, Malvern Instruments Ltd, UK). The equilibrium time chosen was 30 s. The viscosity of water was set to 1.0031 mPa s, and the refractive index was set to 1.330 for the calculation in the application software. A mean value was calculated from more than six measurements; during each measurement, zeta potential was measured for three 20 s runs without any interval between runs.

As one of the common methods to obtain an average particle diameter, hydrodynamic diameter ($d_H$) of AgNPs was measured by Zetasizer equipped with the DLS technique [34].

The relationship between the speed of Brownian motion [35] of a particle and that particle's $d_H$ is defined by the Stokes–Einstein equation [35,36],

$$D = \frac{k_B T}{3\pi\eta d_H},\tag{3.1}$$

where $D$ is the diffusion coefficient, $k_B$ is the Boltzmann constant, $\eta$ is the dynamic viscosity and $T$ is the absolute temperature.

The resistance was measured with a digital multimeter for determining the conductivity of ink spots manually dropped on ink-jet paper and subsequently sintered at different temperatures.

# 4. Results and discussion

## 4.1. Characterization of water in oil nanoemulsion

Our results show that an increase in the ratio of Tween 80 leads to an increase, then a decrease of the maximum emulsifiable water content (figure 2). Consequently, the highest water content of 5.12 wt% on the total blend was attained at a ratio of Tween 80 : Span 80 = 50 : 50.

Studies were carried out using the above ratio of Tween 80 : Span 80 to determine zones where nanoemulsions were formed. It is already established that the mixture becomes a W/O nanoemulsion with a high paraffin content, high surfactant blend content and low water content.

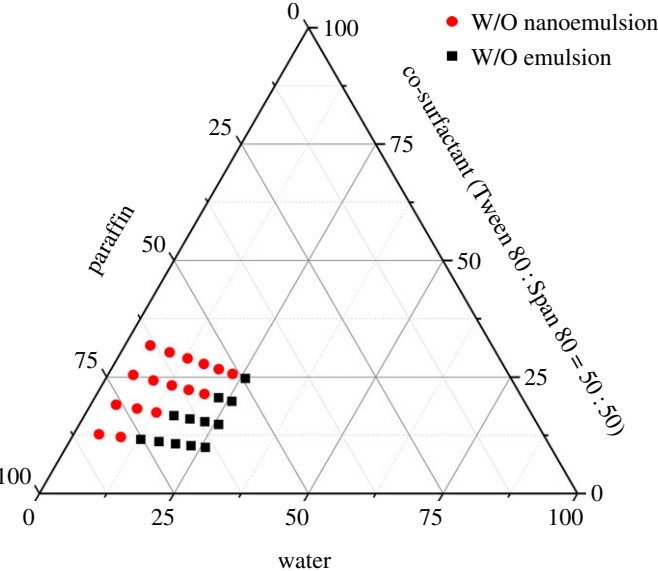

**Figure 3.** Existence regions of W/O emulsion and W/O nanoemulsion.

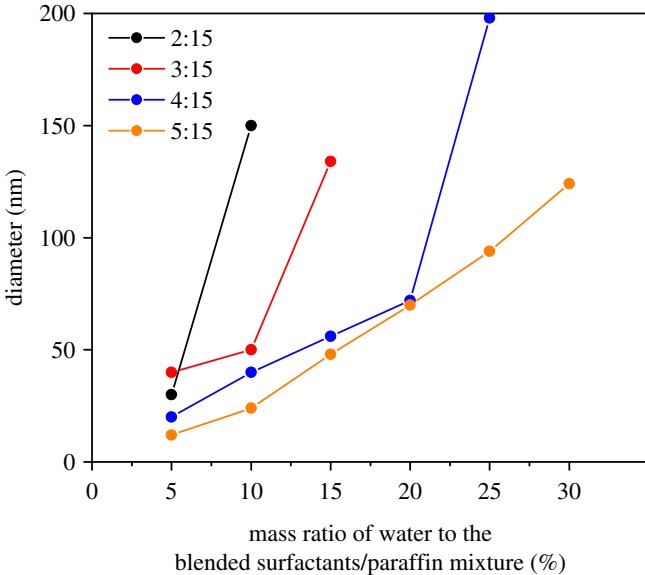

**Figure 4.** Droplet diameter as a function of water content on surfactant blend-paraffin mixture of emulsions.

Meanwhile, with an increase in the water phase in the system, the emulsion transforms from a W/O nanoemulsion to a W/O emulsion and finally becomes an O/W emulsion, causing water droplets to grow in diameter, as predicted according to figure 3. Namely, the emulsion system with larger amounts of surfactant is more likely to stably form a W/O nanoemulsion with smaller water droplets and has the capability to hold more water, if added, despite decreasing stability and shifting to W/O emulsion with larger water droplets. As another variable, if a $AgNO_3$ aqueous solution at as high a concentration as possible is contained instead of only water, the total amount of silver in the emulsion accordingly reaches a maximum, as is expected to provide a silver nano-ink with a maximum content of AgNPs.

Furthermore, as shown in figure 4, the water droplet diameters of the nanoemulsion measured by DLS increased systematically with an increase in the water content of the emulsion. With the constant ratio of water content, nanoemulsion with higher surfactant tends to have smaller droplet diameters. As mentioned above, the diameters of AgNPs can be controlled by adjusting the ratio between oil, surfactant and $AgNO_3$ aqueous solution.

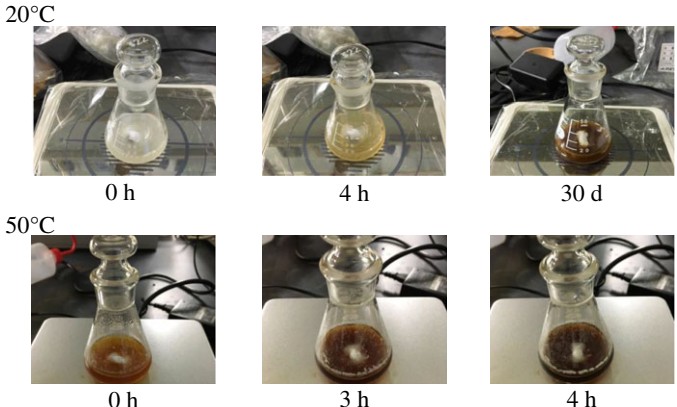

20°C

0 h 4 h 30 d

50°C

0 h 3 h 4 h

**Figure 5.** Reaction speed related to the temperature.

## 4.2. Characterization of silver nanoparticles

During heating, the colour of the solution turned dark brown. In order to understand which components are involved in the chemical change, every possible combination of the components contained in the AgNP nanoemulsion was mixed and heated. The result showed that chemical change only occurred using the $AgNO_3$ aqueous solution and Tween 80.

The speed of the reaction increased with an increase in temperature (figure 5); however, a stable condition with no chemical change was maintained at room temperature or lower for a long period of time.

The Gibbs free energy is expressed as

$$\Delta G = \Delta H - T\Delta S, \tag{4.1}$$

where $\Delta G$ is the Gibbs free energy, $\Delta H$ is the enthalpy change, $T$ is the absolute temperature and $\Delta S$ is the entropy change.

During the reaction, a solid phase is produced from the liquid phase, which indicates that $\Delta S < 0$. As the reaction occurs spontaneously ($\Delta G < 0$), $\Delta H$ is negative with a large absolute value. Furthermore, this reaction occurs spontaneously and slowly at a relatively low temperature.

Figure 6 shows the XRD patterns of the solid AgNPs prepared from an AgNP nanoemulsion containing 6 g paraffin, 1 g Span 80, 1 g Tween 80 and 2 ml aqueous 0.30 g ml$^{-1}$ $AgNO_3$ solutions, which was then purified and sintered at 80°C.

The sample patterns show characteristic signals of silver crystals at 38.1° (111), 44.1° (200), 64.4° (220), 77.3° (311) and 81.3° (222) corresponding to Ag crystals (JCPDS, File No. 4-0783). There were no peaks of crystal impurities, indicating a high purity of synthesized AgNPs.

Figure 7 shows TG curves of Tween 80, Span 80 and the $H_2O$/paraffin nanoemulsion containing 2 g $H_2O$, 6 g paraffin, 1 g Span 80 and 1 g Tween 80. For the $H_2O$/paraffin nanoemulsion, a weight loss period at approximately 100°C is attributed to water evaporation, followed by the volatilization of paraffin at approximately 120–205°C with only one step. The surfactants lost weight at 350–450°C, depending on the type of surfactant, compared with the TG curves of Tween 80 and Span 80 [37].

Figure 8 shows a TG curve of dried AgNPs, that exhibits a similar degradation tendency to the $H_2O$/ paraffin nanoemulsion at 40–500°C, losing 1.97% in weight. The degradation peak at 158°C is related to the residual functional groups from Tween 80, which can also be observed in the DTG curve with a rapid weight loss (15.8 mg min$^{-1}$) as shown in electronic supplementary material, figure S1. For later heating period from 500–900°C, the weight of dried AgNPs presented a similar decreasing speed, but a rapid decrease at approximately 960–968°C. This degradation period in the TG and DTG curves corresponds to the melting point of $Ag^0$, where DTA shows a sharp endothermic peak.

This sharp peak has a reversal that means a momentary weight gain (only 0.09% in weight), then a very sharp decrease due to explosive decomposition with the recoil effect. Some references also showed a similar phenomenon, where Muhamad *et al.* applied two weight gain peaks for eutectic and monotectic reactions (melting) temperatures for the mixture of Ag and CuO [38], while Tsuzuki and McCormick did not point out a tiny reversal peak for ZnO nanoparticles [39]. Jagadeesh Babu *et al.* linked the sharp exotherm peak to the decomposition/dissociation of the salt as free temozolomide [40]. Although the TG curve rapidly changes, there is only one endothermic peak in the DTA curve (91.3 µV s mg$^{-1}$ at

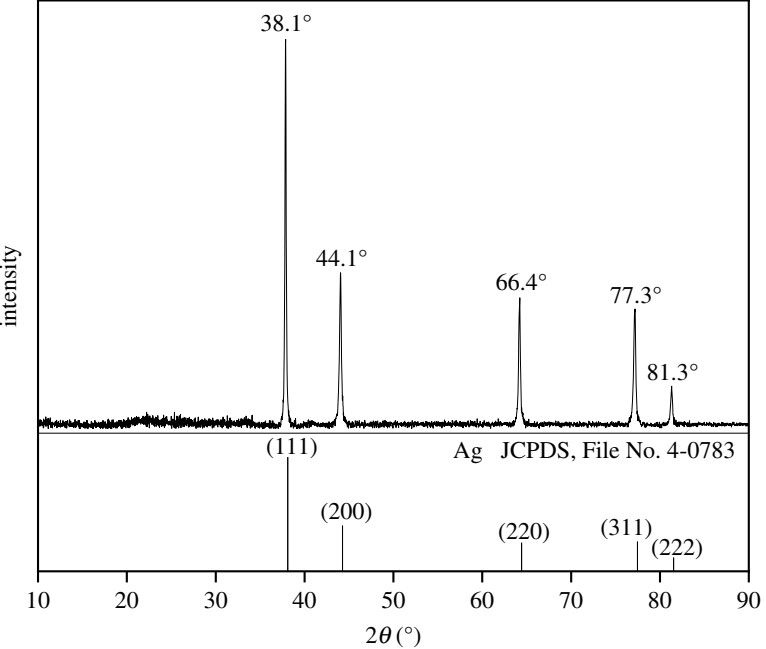

**Figure 6.** X-ray diffraction pattern of produced AgNPs.

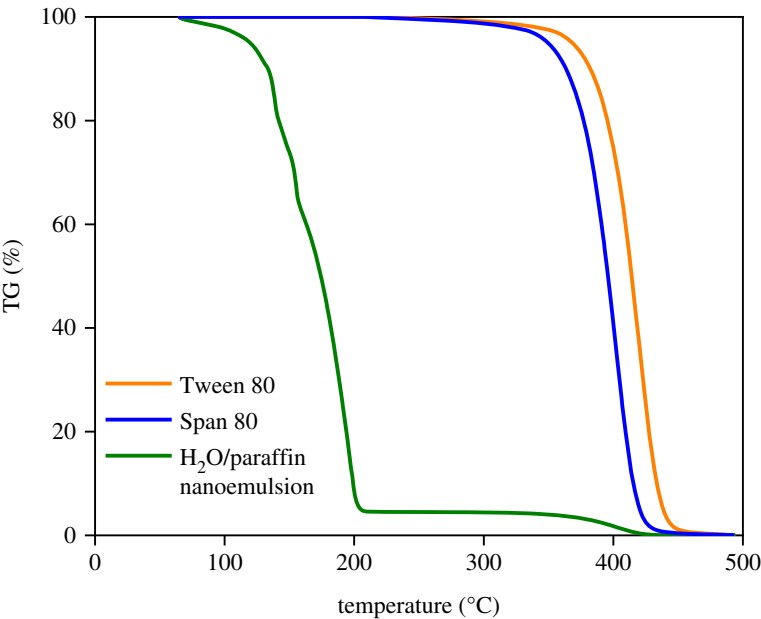

**Figure 7.** Thermogravimetry curves of Tween 80, Span 80 and H₂O/paraffin nanoemulsion.

961°C), suggesting a decomposition reaction of AgO, slightly contained in AgNPs, releasing oxygen during the melting process of $Ag^0$.

In addition, the total loss of weight was approximately 3.4%, corresponding to the surfactant component residual in AgNPs, which indicates a high concentration (purity of 96.6%) of $Ag^0$ in the final AgNP product.

In the FTIR spectra of Tween 80 shown in figure 9, an O–H peak appears at approximately 3439 cm$^{-1}$, and the –CH$_2$– stretching vibrations are present at approximately 2932 and 2866 cm$^{-1}$ (asymmetrical stretch and symmetrical stretch, respectively) [41]. The sharp and symmetric characteristic peak at 1746 cm$^{-1}$ is presumably due to the stretching vibration of C=O in an ester carbonyl group [42,43]. The peak at approximately 1647 cm$^{-1}$ is regarded as a stretching vibration of

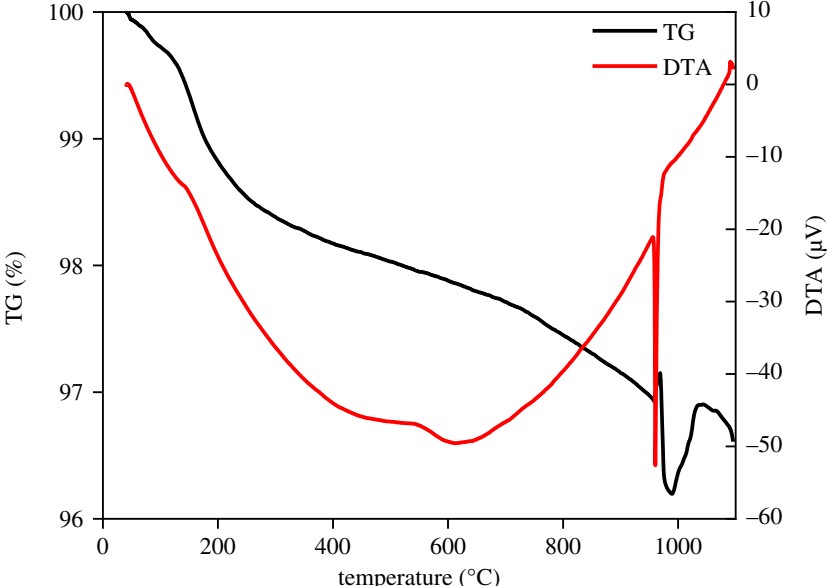

**Figure 8.** Thermogravimetry/differential thermal analysis curves of solid AgNPs.

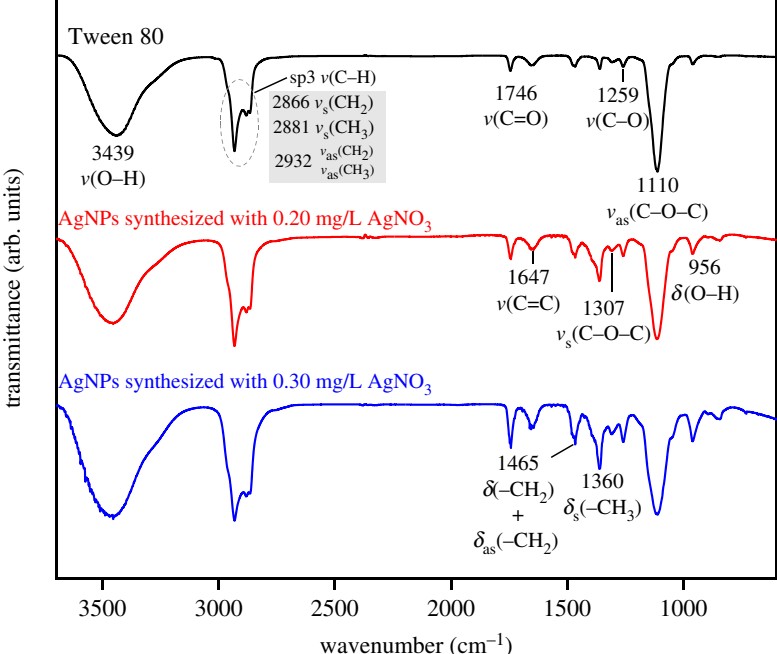

**Figure 9.** FTIR spectra of AgNPs prepared without purification from emulsions with different [AgNO$_3$] mixed with Tween 80.

C=C [42,44]. The peak at 1465 cm$^{-1}$ is mainly attributed to the bending vibrations of –CH$_2$, although the asymmetrical bending vibrations of –CH$_3$ are involved as well [41,45]. The 1360 cm$^{-1}$ peak is attributed to the symmetrical bending vibrations of –CH$_3$ [46]. The C–O peak of appears at 1307 cm$^{-1}$ (symmetrical stretch) and 1110 cm$^{-1}$ (asymmetrical stretch), respectively, while the stretching vibration of C–O is visible at 1259 cm$^{-1}$ [47–49].

Based on comparison with the FTIR results for Tween 80, several functional groups, such as –CH$_3$, C=C and C=O, are found to be present in the purified solid AgNPs.

Furthermore, the transmittance of C–O–C decreased after mixing Tween 80 and AgNO$_3$, while that of C=O increased with increased [AgNO$_3$] in the emulsion system.

Some studies have demonstrated that surfactants with specific functional groups could reduce metal ions into metal via redox reaction [50–54]. For example, free radicals of the α-carbon of ether oxygen in

**Figure 10.** Mechanism of redox reaction between Tween 80 and Ag$^+$.

nonionic polyoxyethylene surfactants (Tween series surfactants) are able to reduce Ag$^+$ into Ag$^0$ as shown in figure 10 [54,55].

In Tween 80 with a chemical structure shown by four sets of parentheses marked with symbols from 'w' to 'z', there are several -R-O-CH$_2$-R'- groups. Namely, it is believed that Tween 80 undergoes a redox reaction, in which a free radical is formed from the α-carbon of the ether oxygen via hydrogen abstraction in the initial stage [54]. Afterwards, the radicals can be further oxidised into esters, or degraded to form aldehydes.

TEM images of AgNPs prepared from AgNP nanoemulsions with different [AgNO$_3$] are shown in figure 11. The morphology of AgNPs was found to be spherical. The $d_S$ was estimated by ImageJ software and found to vary as 8.6, 8.8, 10.1 and 13.4 nm with varying [AgNO$_3$] of 0.10, 0.20, 0.30 and 0.50 g ml$^{-1}$, respectively. These findings prove that the particle diameter of the synthesized AgNPs increases with an increase in [AgNO$_3$] of the AgNPs nanoemulsions.

When the number of nuclei remained constant, or increased slower than the total ions due to the high Ag$^+$ ion concentration, the atoms formed at the latter period were used for the growth of particles and resulted in the formation of larger particles [56,57]. In addition, those particles had a smaller diameter than the droplets of the H$_2$O nanoemulsion itself (approx. 73 nm), as presented in figure 4. This finding also indicates that this nanoemulsion system could provide an extremely tiny microreactor to produce nanomaterials, especially by reduction.

The $d_H$ was found to be 16.9, 18.4, 20.4 and 24.2 nm with different [AgNO$_3$] of 0.10, 0.20, 0.30 and 0.50 g ml$^{-1}$, as shown in electronic supplementary material, table S1. Compared with the $d_S$ results obtained by TEM, the average $d_H$ measured by Zetasizer showed almost two times.

In this study, the synthesized AgNPs have been confirmed with some functional groups stabilized on the surface, which provides extra shells to the AgNPs, leading to a larger $d_H$ through DLS method than the $d_S$ through TEM [58], as shown in electronic supplementary material, figure S2, and similar results have been reported in the previous studies [34,59,60].

Figure 12 shows the UV-visible spectra of the AgNP dispersions synthesized with different concentrations of AgNO$_3$. A red shift from 407 to 415 nm occurred with an increased [AgNO$_3$]. According to Mie's theory [61], this phenomenon indicates the crystallization of AgNPs with increased diameter because of the increased initial [AgNO$_3$]. However, the negative absorbance values should be related to the instability of the AgNP solution, or the C=O conjugation that exists on the surfaces of the particles generated from surfactant groups after the reaction.

For a stable colloid system, surface effects play a more important role than bulk effects. As for the most accepted Derjaguin–Landau–Verwey–Overbeek (DLVO) theory [62,63], the stability of a colloid can be reflected by the sum of van der Waals attractive forces and electrostatic repulsive forces within the electrical double layer model [64]. The zeta potential reflects the stability by electrostatic repulsion, as it measures the effective charge of the particle surface [65]. In general, nanoparticles with absolute

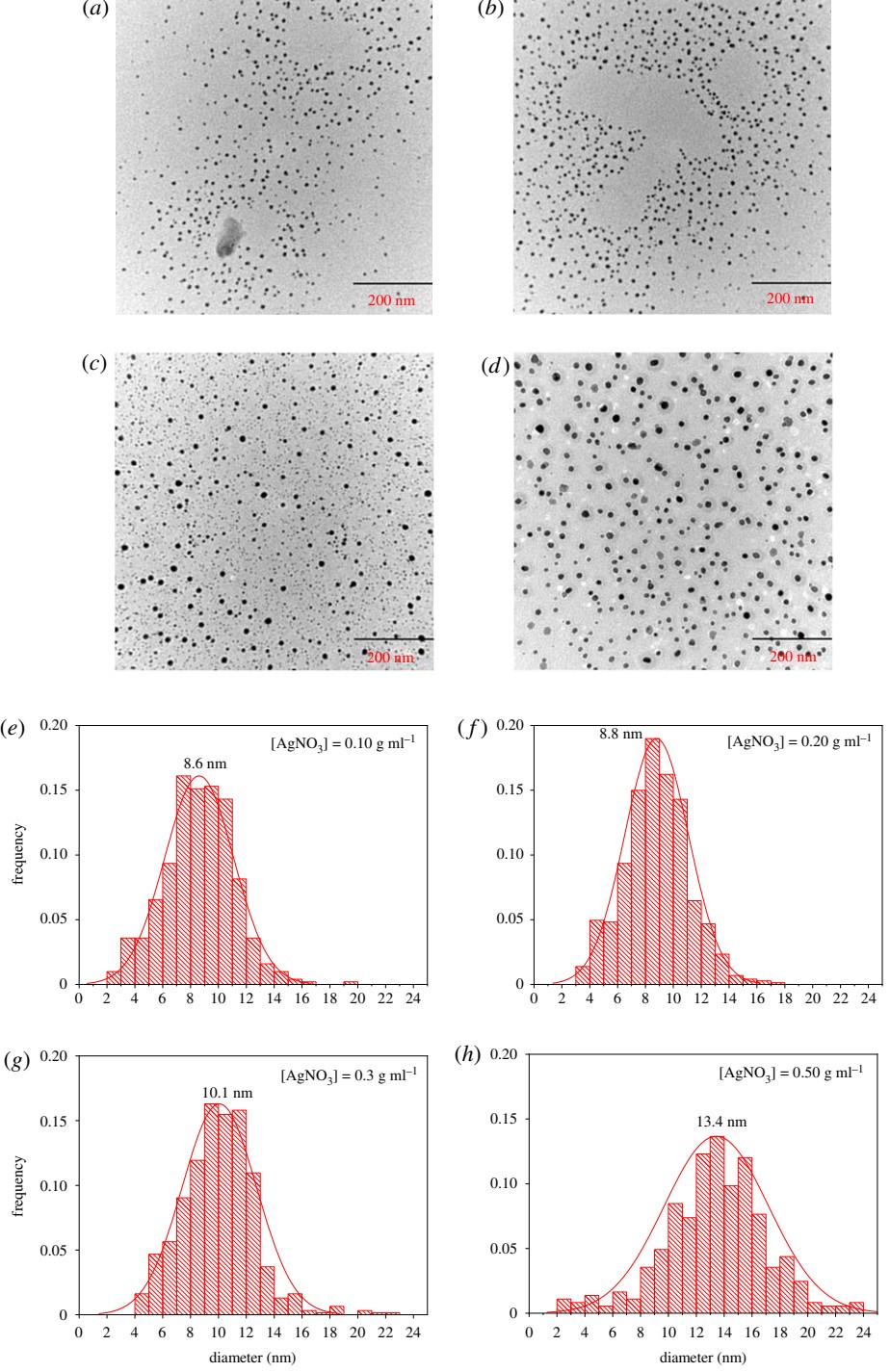

**Figure 11.** TEM images and histogram of particle size distribution for solid AgNPs synthesized from AgNP nanoemulsions with different [AgNO₃] of 0.10 g ml⁻¹ (*a* and *e*), 0.20 g ml⁻¹ (*b* and *f*), 0.30 g ml⁻¹ (*c* and *g*) and 0.50 g ml⁻¹ (*d* and *h*).

zeta potential values greater than 30 mV have high degrees of stability [66]. Figure 13 shows that all the AgNP samples had highly negative zeta potentials varying from −45.8 to −50.6 mV, implying that they were all stably dispersed. The high absolute zeta potentials ensure the high dispersion stability of the AgNP ink, due to the residual functional groups (–OH, –O–C=O, etc.) from Tween 80 present in the product (FTIR). These functional groups provide electrosteric repulsion, which is the combination of electrostatic repulsion arising from the negative charge condition of the mentioned organic functional groups on the AgNP surface and steric repulsion derived from the long-chain molecules on the AgNP surface. The synthesized AgNP inks practically maintained a stable condition, even after long-term

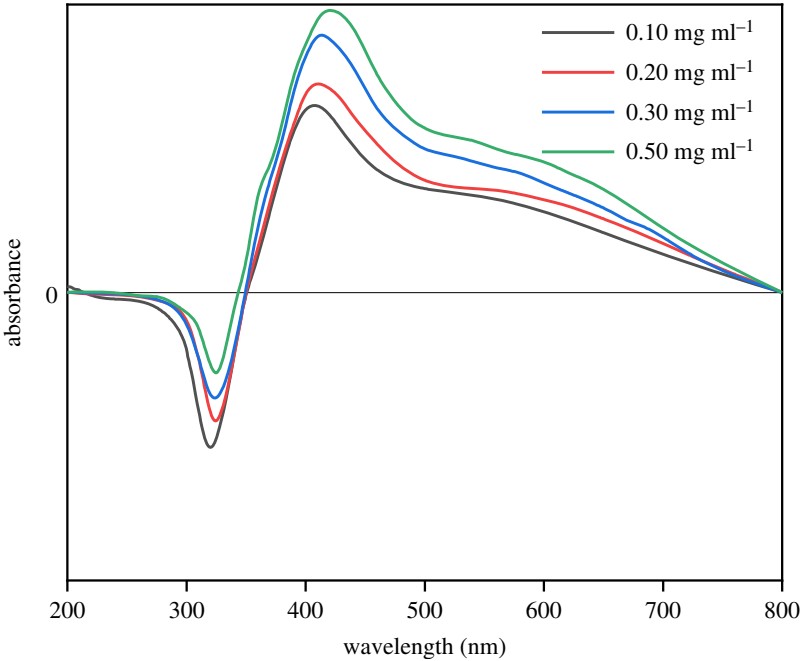

**Figure 12.** UV-visible spectra of AgNPs synthesized from AgNO$_3$ nanoemulsion with different [AgNO$_3$].

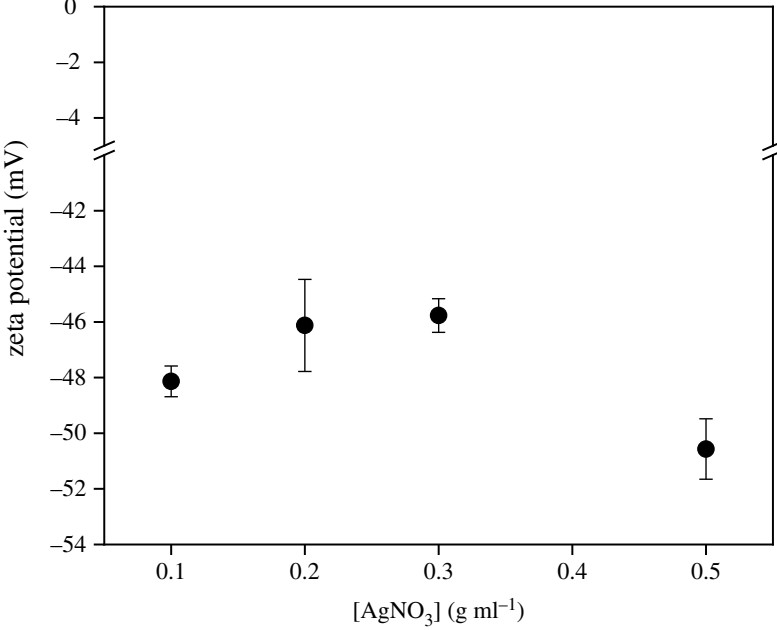

**Figure 13.** Zeta potential of synthesized AgNPs with different [AgNO$_3$].

storage (five months) at room temperature, with only slight sedimentation (figure 14). Therefore, the negative values appearing in the UV-visible spectrum are probably not caused by the instability of the AgNP solution, but mainly resulted from the C=O conjugation.

As shown in table 1, the electrode created by drying the AgNP nanoemulsion at room temperature had high resistance; however, it became conductive after sintering the nanoemulsion at approximately 50°C, which accelerates the reduction of Ag$^+$ to solid Ag$^0$. The sintering temperature should be set as high as possible for higher conductivity, but in a reasonable range that causes no damage to the substrate. The agglomerated AgNPs obtained by centrifugation of a nanoemulsion after heating at 60°C showed excellent conductivity (0 Ω).

(*a*)    [AgNO$_3$] = 0.10 g ml$^{-1}$    [AgNO$_3$] = 0.20 g ml$^{-1}$    [AgNO$_3$] = 0.30 g ml$^{-1}$    [AgNO$_3$] = 0.50 g ml$^{-1}$

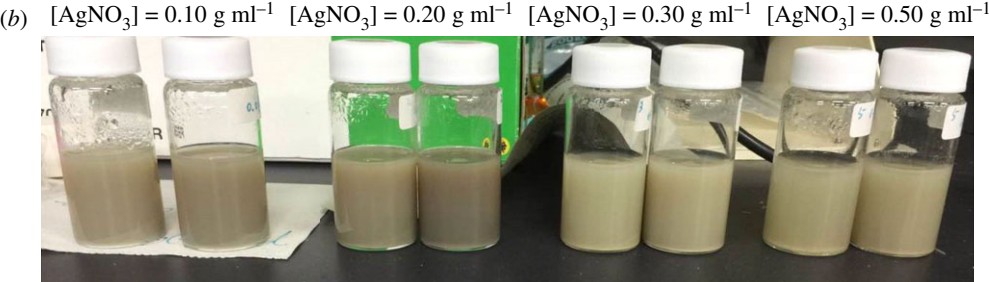

(*b*)    [AgNO$_3$] = 0.10 g ml$^{-1}$    [AgNO$_3$] = 0.20 g ml$^{-1}$    [AgNO$_3$] = 0.30 g ml$^{-1}$    [AgNO$_3$] = 0.50 g ml$^{-1}$

**Figure 14.** Sedimentation condition of AgNPs one week (*a*) and five months (*b*) after synthesis.

**Table 1.** Resistance of AgNP deposits on ink-jet paper after sintering at different temperatures and an agglomerate of AgNPs by centrifugation from sintered AgNPs.

| form | sintering temperature (℃) | resistance ($\Omega$) |
|---|---|---|
| deposit on ink-jet paper | 25 | $20 \times 10^6$–$30 \times 10^6$ |
| | 60 | 5–45 |
| | 90 | 0–30 |
| | 150 | 0–20 |
| | 250 | 0–15 |
| agglomerate | 60 | 0 |

## 5. Conclusion

A self-assembled AgNP nanoemulsion ink was prepared successfully through a one-step nanoemulsion method, where Tween 80 can act not only as a surfactant, but also as a reducing agent to obtain AgNPs, and as a stabilizer to prevent aggregation after the reduction. The mechanism of the reaction between Tween 80 and AgNO$_3$ is proven to be a redox reaction caused by a free radical of an α-carbon of ether oxygen and Ag$^+$. The ink remains as a nanoemulsion at low temperature; however, it produces AgNPs with diameters of 8.6–13.4 nm after a sintering procedure, ensuring a high conductivity and good stability. Regulation of the particle size is feasible, and smaller particles are easily obtained by adjusting certain factors: (i) water content, (ii) surfactant-to-paraffin ratio, and (iii) AgNO$_3$ concentration. The AgNP inks had high absolute values of zeta potential varying from −45.8 to −50.6 mV, implying that they were all stably dispersed, although there was no significant difference in [AgNO$_3$]. The high stability of the ink proves that the high absorbance values in the UV-visible spectrum are related to C=O conjugation.

Data accessibility. All datasets and codes have been uploaded in the Dryad Digital Repository: https://doi.org/10.5061/dryad.jwstqjq5x [67].
Authors' contributions. D.H. performed most of the laboratory work. K.O. and M.K guided and assisted in part of the experiments. D.H. and T.E. contributed to the design of the study and drafted the manuscript. T.E. supervised the project. All authors have given final approval to publish the article.
Competing interests. There are no conflicts to declare.

Funding. This work was supported by JSPS KAKENHI grant no. 17KT0069 and the operating budget of University of Tsukuba for language editing and the article processing charge for an open access article.

Acknowledgements. The authors would like to thank the open facility centre for science and technology, University of Tsukuba for allowing us to use their facilities.

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
