## [Reviewer comments · Royal Society Open Science]

Review History

RSOS-200296.R0 (Original submission)

Review form: Reviewer 1

Is the manuscript scientifically sound in its present form?

Yes

Are the interpretations and conclusions justified by the results?

Yes

Is the language acceptable?

No

Do you have any ethical concerns with this paper?

No

Have you any concerns about statistical analyses in this paper?

Yes

Recommendation?

Accept with minor revision (please list in comments)

Comments to the Author(s)

The detailed comments are as follows:

- (1) There are some grammar errors in the text. Please carefully modify the full text.
- (2) At line 46 Synthesis and purification of AgNPs A mean value was calculated from four to five measurements. Are you calculated SD or SE for your samples.
- (3) At line 56 in Synthesis and purification of AgNPs (conducted thrice) it's better to use three times and modify the sentences: Purified solid AgNPs were obtained by a subsequent purification procedure with ethanol (conducted thrice) to remove surfactants and paraffin.
- (4) Results and discussion Characterisation of W/O nanoemulsion at line 18 Namely, the nanoemulsion with higher amounts of surfactant holds more water, and thus, the droplet size is smaller due to the emulsification characteristics. Such a nanoemulsion-based ink has a potential to maintain the maximum content of silver nitrate aqueous solution, which is expected to provide higher contents of AgNPs. It's confusing ??? how the higher concentration of Ag⁺ increases the size of AgNP and decrease it at the same time, please clarify it.
- (5) Thermodynamic at line 2 in Results and discussion need more explanation.
- (6) Supplementary material it's only one figure I think it will be more convenient if you insert it in the manuscript if you choose to add the figure as SM no mater but, Please include the following information in the file: article title, author names; affiliation and e-mail address of the corresponding author.
- (7) Finally, Did you try to utilize Ag NP ink in the fabrication of electrochemical biosensor for sensing real samples such as drugs, pesticidesext.

Review form: Reviewer 2

Is the manuscript scientifically sound in its present form?

Yes

Are the interpretations and conclusions justified by the results?

Yes

Is the language acceptable?

Yes

Do you have any ethical concerns with this paper?

No

Have you any concerns about statistical analyses in this paper?

No

Recommendation?

Accept as is

Comments to the Author(s)

Good as such. Please fix the following minor issues:

P1L58 - Fig 2 should be Fig 1?

P2L28 - 20:80 should be 80:20?

P2L36 - should the particle size distribution be called droplet size distribution?

P5L14 second paragraph: zeta potential has excessive accuracy of 0.01mV.

Review form: Reviewer 3

Is the manuscript scientifically sound in its present form?

No

Are the interpretations and conclusions justified by the results?

Yes

Is the language acceptable?

Yes

Do you have any ethical concerns with this paper?

No

Have you any concerns about statistical analyses in this paper?

No

Recommendation?

Major revision is needed (please make suggestions in comments)

Comments to the Author(s)

A revision file has been attached herewith (Appendix A).

Decision letter (RSOS-200296.R0)

03-Apr-2020

Dear Dr Enomae:

Title: Characterization of Self-Assembled Silver Nanoparticle Ink Based on Nanoemulsion

Method

Manuscript ID: RSOS-200296

The editor assigned to your manuscript has now received comments from reviewers. We would like you to revise your paper in accordance with the referee and Subject Editor suggestions which can be found below (not including confidential reports to the Editor). Please note this decision does not guarantee eventual acceptance.

Please submit your revised paper before 26-Apr-2020. Please note that the revision deadline will expire at 00.00am on this date. If we do not hear from you within this time then it will be assumed that the paper has been withdrawn. In exceptional circumstances, extensions may be possible if agreed with the Editorial Office in advance. We do not allow multiple rounds of revision so we urge you to make every effort to fully address all of the comments at this stage. If deemed necessary by the Editors, your manuscript will be sent back to one or more of the original reviewers for assessment. If the original reviewers are not available we may invite new reviewers.

To revise your manuscript, log into <http://mc.manuscriptcentral.com/rsos> and enter your Author Centre, where you will find your manuscript title listed under "Manuscripts with

Decisions." Under "Actions," click on "Create a Revision." Your manuscript number has been appended to denote a revision. Revise your manuscript and upload a new version through your Author Centre.

RSC Associate Editor:
Comments to the Author:
(There are no comments.)

RSC Subject Editor:
Comments to the Author:
(There are no comments.)

Reviewers' Comments to Author:
Reviewer: 1

Comments to the Author(s)

The detailed comments are as follows:

- (1) There are some grammar errors in the text. Please carefully modify the full text.
- (2) At line 46 Synthesis and purification of AgNPs A mean value was calculated from four to five measurements. Are you calculated SD or SE for your samples.
- (3) At line 56 in Synthesis and purification of AgNPs (conducted thrice) it's better to use three times and modify the sentences: Purified solid AgNPs were obtained by a subsequent purification procedure with ethanol (conducted thrice) to remove surfactants and paraffin.
- (4) Results and discussion Characterisation of W/O nanoemulsion at line 18 Namely, the nanoemulsion with higher amounts of surfactant holds more water, and thus, the droplet size is smaller due to the emulsification characteristics. Such a nanoemulsion-based ink has a potential to maintain the maximum content of silver nitrate aqueous solution, which is expected to provide higher contents of AgNPs. It's confusing ??? how the higher concentration of Ag⁺ increases the size of AgNP and decrease it at the same time, please clarify it.

(5) Thermodynamic at line 2 in Results and discussion need more explanation.

(6) Supplementary material it's only one figure I think it will be more convenient if you insert it in the manuscript if you choose to add the figure as SM no mater but, Please include the following information in the file: article title, author names; affiliation and e-mail address of the corresponding author.

(7) Finally, Did you try to utilize Ag NP ink in the fabrication of electrochemical biosensor for sensing real samples such as drugs, pesticidesext.

Reviewer: 2

Comments to the Author(s)

Good as such. Please fix the following minor issues:

P1L58 - Fig 2 should be Fig 1?

P2L28 - 20:80 should be 80:20?

P2L36 - should the particle size distribution be called droplet size distribution?

P5L14 second paragraph: zeta potential has excessive accuracy of 0.01mV.

Reviewer: 3

Comments to the Author(s)

A revision file has been attached herewith.

Author's Response to Decision Letter for (RSOS-200296.R0)

See Appendix B.

Decision letter (RSOS-200296.R1)

Dear Dr Enomae:

Title: Characterisation of Self-Assembled Silver Nanoparticle Ink Based on Nanoemulsion Method

Manuscript ID: RSOS-200296.R1

It is a pleasure to accept your manuscript in its current form for publication in Royal Society Open Science. The chemistry content of Royal Society Open Science is published in collaboration with the Royal Society of Chemistry.

RSC Associate Editor
Comments to the Author:
(There are no comments.)

Reviewer(s)' Comments to Author:

Appendix A

The article under review “Characterization of Self-Assembled Silver Nanoparticle Ink Based on Nanoemulsion Method” presents the promising results with good description and may be accepted for publication after addressing the following queries. (Manuscript ID: RSOS-200296)

Comment #1

For the discussion of Figure 6 (TG-discussion),

It would be better if authors can also discuss the heat capacity of the subjected material.

Comment #2

For the FTIR discussion:

Kindly justify the following findings with a valid reported reference;

- 1 The sharp and symmetric characteristic peak at 1,746 cm⁻¹ is presumably due to the stretching vibration of C=O in an ester carbonyl group (Reference ?).
- 2 The peak at approximately 1,647 cm⁻¹ is regarded as a stretching vibration of C=C (Reference?).
- 3 The peak at 1,465 cm⁻¹ is mainly attributed to the bending vibrations of –CH₂, although the asymmetrical bending vibrations of –CH₃ are involved as well. The 1,360 cm⁻¹ peak is attributed to the symmetrical bending vibrations of –CH₃. The C-O-C peak of the ester group appears at 1,307 cm⁻¹ (symmetrical stretch), and 1,259 cm⁻¹ (symmetrical stretch) respectively, while the stretching variation of C-O-C is visible at 1,110 cm⁻¹ (Reference ?).
- 4 Why transmittance decreased for C-O-C while increased for C=O?
- 5 FTIR spectra should be revised and scale should be from left to right.

Comment #3

Kindly label all the diffraction peaks in Figure 5 (XRD results) and please cite the reference.

Comment # 4

In the TEM description, it is written that the particles are spherical in shape with varying sizes from 8.6 to 13.4 nm depending upon the concentration of AgNO₃. As TEM is a powerful tool to study the morphology of nanoparticles, therefore it is better if you could provide high magnification images as inset in Figure 10 of the prepared samples which could prove your statement regarding the size and shape of the nanoparticles.

Comment # 5

TGA analysis revealed only about 3.3% weight loss at around 960 °C and then weight was increased suddenly. Why? Explain TGA/DTA in detail please.

Comment # 6

Why there is red shift occurred from 407 to 415 nm due to increased AgNO₃?

Comment # 7

Why authors choose emulsion method to prepare AgNPs rather than green method using plant extract? Emulsion is an old technique to synthesize nanoparticles. Mention the novelty of the chosen method.

Comment # 8

Add more detail on zeta size measurements.

Comment # 9

Improve the visibility of all Figures.

Comment # 10

Almost all characterization results are superficially explained. It is a need to explain in more detail with strong evidence.

Appendix B

Reviewer: 1

Comments to the Author(s)

The detailed comments are as follows:

(1) There are some grammar errors in the text. Please carefully modify the full text.

We really appreciate your useful comments. We have checked the grammar errors and corrected some errors in the manuscript:

Our manuscript has been already language edited before the submission. If you still find grammar errors, We would be pleased to correct them following your indication.

(2) At line 46 Synthesis and purification of AgNPs A mean value was calculated from four to five measurements. Are you calculated SD or SE for your samples.

We appreciate your pointing out this shortage of data. We actually applied at least six samples to measure zeta potential (we have revised the information in the manuscript: A mean value was calculated from more than six measurements.). So, I added new Table 1 below:

Table S2 Zeta potential of synthesised AgNPs with different [AgNO₃]

[AgNO ₃] (g/mL)	Zeta Potential (mV)	SD (mV)	Number of samples measured
0.10	-48.1	1.1	9
0.20	-46.1	3.3	13
0.30	-45.8	1.2	15
0.50	-50.6	2.2	6

(3) At line 56 in Synthesis and purification of AgNPs (conducted thrice) it's better to use three times and modify the sentences: Purified solid AgNPs were obtained by a subsequent purification procedure with ethanol (conducted thrice) to remove surfactants and paraffin.

We appreciate your comment, and we have changed the sentence as below in the body text as:

“Purified solid AgNPs were obtained by a subsequent purification procedure with ethanol **three times** to remove surfactants and paraffin.”

(4) Results and discussion Characterisation of W/O nanoemulsion at line 18 Namely, the nanoemulsion with higher amounts of surfactant holds more water, and thus, the droplet size is smaller due to the emulsification characteristics. Such a nanoemulsion-based ink has a potential to maintain the maximum content of silver nitrate aqueous solution, which

is expected to provide higher contents of AgNPs. It's confusing ??? how the higher concentration of Ag⁺ increases the size of AgNP and decrease it at the same time, please clarify it.

We appreciate your question and apologize for the unclear explanation.

To explain the water content and droplet diameter conditions of the nanoemulsion clearly, we would like to discuss them one by one (water content firstly, and then diameters). The revised text is:

“Namely, the emulsion system with larger amounts of surfactant is more likely to stably form a W/O nanoemulsion with smaller water droplets and has the capability to hold more water, if added, despite decreasing stability and shifting to W/O emulsion with larger water droplets. As another variable, if a AgNO₃ aqueous solution at as a high concentration as possible is contained instead of only water, the total amount of silver in the emulsion accordingly reaches a maximum, as is expected to provide a silver nano-ink with a maximum content of AgNPs.

Furthermore, as shown in Fig. 4, the water droplet diameters of the nanoemulsion measured by DLS increased systematically with an increase in the water content of the emulsion. With the constant ratio of water content, nanoemulsion with higher surfactant tends to have smaller droplet diameters. As mentioned above, the diameters of AgNPs can be controlled by adjusting the ratio between oil, surfactant, and AgNO₃ aqueous solution.”

(5) Thermodynamic at line 2 in Results and discussion need more explanation.

We do not know which part the reviewer referred to for this comment. We would be pleased to have the reviewer point it out again.

(6) Supplementary material it's only one figure I think it will be more convenient if you insert it in the manuscript if you choose to add the figure as SM no mater but, Please include the following information in the file: article title, author names; affiliation and e-mail address of the corresponding author.

We appreciate your suggestion for the supplemental material. We have moved the former SM-Fig. 1 to the main text body as Fig. 4..

Besides, we will add the article title and author information in SM part in the final publication after the review process is completed.

(7) Finally, Did you try to utilize Ag NP ink in the fabrication of electrochemical biosensor for sensing real samples such as drugs, pesticidesext.

We appreciate your suggestion. Unfortunately, we have not apply this ink for the sensor application in this study. The application of the AgNP ink will be included in a later article.

Reviewer: 2

Comments to the Author(s)

Good as such. Please fix the following minor issues:

P1L58 - Fig 2 should be Fig 1?

We appreciate your reminding, and we have corrected the wrong numbers of all figures.

P2L28 - 20:80 should be 80:20?

According to your suggestion, I replaced "20:80" with "80:20".

P2L36 - should the particle size distribution be called droplet size distribution?

We appreciate your comment here, and we do think that we should use "droplet size distribution" as you offered the suggestion to us.

P5L14 second paragraph: zeta potential has excessive accuracy of 0.01mV.

The accuracy of the original data was 0.1 mV. However, the zeta potentials used in the text were the mean values after several times of measurement, and we applied 0.1 mV as the accuracy.

We appreciate your comment about the accuracy. We modified all the values to an accuracy of 0.1 mV.

Reviewer 3

The article under review "Characterization of Self-Assembled Silver Nanoparticle Ink Based on Nanoemulsion Method" presents the promising results with good description and may be accepted for publication after addressing the following queries. (Manuscript ID: RSOS-200296)

Comment #1

For the discussion of Figure 6 (TG-discussion),
It would be better if authors can also discuss the heat capacity of the subjected material.

We appreciate your suggestion. We understand that heat capacity should be discussed. Unfortunately, however, we did not apply DSC (Differential Scanning Calorimetry) for the material, but applied DTA (Differential Thermal-gravimetry Analysis) from which heat flow is not obtained in this study.

Comment #2

For the FTIR discussion:

Kindly justify the following findings with a valid reported reference;

1. The sharp and symmetric characteristic peak at $1,746\text{ cm}^{-1}$ is presumably due to the stretching vibration of C=O in an ester carbonyl group (Reference ?).

We appreciate your suggestion. We have added some references to support the C=O stretching vibration result.

2. The peak at approximately $1,647\text{ cm}^{-1}$ is regarded as a stretching vibration of C=C (Reference?).

We appreciate your suggestion. We have added some references to support the C=C stretching vibration result.

3. The peak at $1,465\text{ cm}^{-1}$ is mainly attributed to the bending vibrations of $-\text{CH}_2$, although the asymmetrical bending vibrations of $-\text{CH}_3$ are involved as well. The $1,360\text{ cm}^{-1}$ peak is attributed to the symmetrical bending vibrations of $-\text{CH}_3$. The C-O-C peak of the ester group appears at $1,307\text{ cm}^{-1}$ (symmetrical stretch), and $1,259\text{ cm}^{-1}$ (symmetrical stretch) respectively, while the stretching variation of C-O-C is visible at $1,110\text{ cm}^{-1}$ (Reference ?).

We appreciate your suggestion. We have added some references to support $-\text{CH}_2-$, -

CH₃, C=O and C-O-C results. Besides the chemical groups mentioned above, we also have added some references to other chemical groups in the main body text.

4. Why transmittance decreased for C-O-C while increased for C=O?

We appreciate your question about the transmittance change. We have mentioned the reason in the main body text as:

(1) We added some references to the redox reaction mechanisms compared between Tween 80 and other Tween series surfactants, followed by discussion on “why transmittance decreased for C-O-C while increased for C=O”.

(2) We have also mentioned that “Afterwards, the radicals can be further oxidized into esters, or degraded to form aldehydes”. Besides, the redox reaction mechanism between Tween 80 and Ag⁺ is also shown in Fig. 9.

It is believed that Tween 80 undergoes oxidation, in which a free radical is formed from the α -carbon of the ether oxygen, and aldehydes groups are formed.

5. FTIR spectra should be revised and scale should be from left to right.

We appreciate your suggestion. We have reversed the scale of the wavenumber.

Comment #3

Kindly label all the diffraction peaks in Figure 5 (XRD results) and please cite the reference.

We appreciate your suggestion. We have revised Figure 5 by adding peak labels, and we have also cited the references in the body text.

Comment # 4

In the TEM description, it is written that the particles are spherical in shape with varying sizes from 8.6 to 13.4 nm depending upon the concentration of AgNO₃. As TEM is a powerful tool to study the morphology of nanoparticles, therefore it is better if you could provide high magnification images as inset in Figure 10 of the prepared samples which could prove your statement regarding the size and shape of the nanoparticles.

We appreciate your suggestion. We have enlarged the figures for readers to better understand the shape and size condition.

Comment # 5

TGA analysis revealed only about 3.3% weight loss at around 960 oC and then weight was increased suddenly. Why? Explain TGA/DTA in detail please.

We appreciate your question. The degradation period at approximately 960 - 968 °C in the TG curve corresponds to the melting point of Ag₂O, where DTA shows a sharp endothermic peak.

To better support the result, we have revised TG-discussion part by adding more information to the text and DTG curve in supporting material, the revised part is:

Fig. 7 shows TG curves of Tween 80, Span 80, and the H₂O/paraffin nanoemulsion containing 2 g H₂O, 6 g paraffin, 1 g Span 80, and 1 g Tween 80. For the H₂O/paraffin nanoemulsion, a weight loss at approximately 100 °C is attributed to water evaporation, followed by the volatilisation of paraffin at approximately 120-205 °C in only one step. The surfactants lost weight at 350 – 450 °C, depending on the type of surfactant, compared with the TGA curves of Tween 80 and Span 80.³⁴

Fig. 8 shows a TG curve of dried AgNPs. It exhibits a similar degradation tendency to the H₂O/paraffin nanoemulsion at 40-500 °C, losing 1.97% weight. The degradation peak at 158 °C is related to the residual chemical groups from Tween 80, which can be also observed in the TGA curve with a rapid weight loss (15.8 mg/min) as shown in Fig. S1. In the higher temperature range at 500-900 °C, the weight of dried AgNPs presented a similar decreasing speed; however a rapid decrease at approximately 960 - 968 °C. This degradation in the TG and DTG curves corresponds to the melting point of Ag₂O, where DTA shows a sharp endothermic peak.

This sharp peak has a reversal that means a momentary weight gain (only 0.09%), then a very sharp decrease due to explosive decomposition with a recoil effect. Some references also showed a similar phenomenon, where Muhamad *et al.* applied two weight gain peaks for eutectic and monotectic reactions (melting) temperatures for the mixture of Ag and CuO,³⁵ while Tsuzuki and McCormick did not point out a tiny reversal peak for ZnO nanoparticles.³⁶ Jagadeesh Babu *et al.* linked the sharp exotherm peak to the decomposition/dissociation of the salt as free temozolomide.³⁷ Although the TG curve rapidly changes, there is only one endothermic peak in DTA curve (91.3 μV.s/mg at 961 °C), suggesting a decomposition reaction of Ag₂O, slightly contained in AgNPs, releasing oxygen during the melting process of Ag₂O.

Fig. S2 DTG curve of solid AgNPs (inner figure is the TGA curve from 40-900 °C).

Comment # 6

Why there is red shift occurred from 407 to 415 nm due to increased AgNO₃?

We appreciate your question about the red shift. We have mentioned that “According to Mie’s theory, this phenomenon indicates the crystallisation of AgNPs with increased diameter because of the increased initial [AgNO₃].” following “A red shift from 407 to 415 nm occurred with an increased [AgNO₃].” We hope this description is satisfactory to explain the red shift occurrence.

Comment # 7

Why authors choose emulsion method to prepare AgNPs rather than green method using plant extract? Emulsion is an old technique to synthesize nanoparticles. Mention the novelty of the chosen method.

We appreciate your suggestion about utilizing plant extract, but the characteristics of our ink are:

(1) High electroconductivity of an ink-jet printed track on paper is realized after sintering at as low as 60 °C (the sintering temperature is very low compared to many other AgNPs), and we can keep the ink as nanoemulsion at low temperature for a long time with high stability.

(2) Tween 80 alone, added to a AgNO₃ solution, plays three roles: a surfactant, reductant, and stabilizer consecutively through the reaction processes.

(3) Regulation of the particle size is feasible, and smaller particles are easily obtained by adjusting several factors: water content, surfactant-to-paraffin ratio, and AgNO₃ concentration.

Plant extract might be fine to fit some of its targets, but not all.

Comment # 8

Add more detail on zeta size measurements.

We appreciate your suggestion, by which we can have readers understand the size distribution better. This study focused on the zeta potential using Zetasizer as the hydrodynamic size measured by the dynamic light scattering technique. In this case, the contribution to the overall signal measured is related to the amount of light scattered by the different particle sizes. The intensity result is the one that comes directly from the data, volume and number are derived, and thus depend on information about the refractive index of the particles (as well as their shape).

The hydrodynamic sizes for very small particles are usually bigger than the size obtained by TEM. We treated this data as incorrect in the previous version, but we have revised by adding the zeta size information in supplementary material with SD, and we also added the information in the text with discussion as:

In the experimental part:

The produced AgNPs were diluted 20 times. A 1.0 mL of the diluted AgNPs was shaken well, poured into a cuvette with a capacity of 1.0 mL, and set to a sample chamber of the instrument (DTS1070, Malvern Instruments Ltd, England). The equilibrium time was chosen 30s. The viscosity of water was set to 1.0031 mPa·s, and the refractive index was set to 1.330 for the calculation in the application software. A mean value was calculated from more than six measurements; during each measurement, zeta potential was measured for 20 s three runs without any interval between runs.

As one of the common methods to obtain an average particle diameter, hydrodynamic diameter (d_H) of AgNPs was measured by Zetasizer equipped with the DLS technique.

³⁴

The relationship between the speed of Brownian motion³⁵ of a particle and that particle's hydrodynamic diameter (d_H) is defined by the Stokes-Einstein equation:^{35, 36}

$$D = \frac{k_B T}{3\pi\eta d_H}, (1)$$

where D is the diffusion coefficient, k_b is the Boltzmann constant, and T is the absolute temperature.

The resistance was measured with a digital multimeter for determining the conductivity of ink spots manually dropped on ink jet paper and subsequently sintered at different temperatures.

In the result part:

The d_H was found to be 16.9, 18.4, 20.4, and 24.2 nm with different $[AgNO_3]$ of 0.10, 0.20, 0.30 and 0.50 g/mL, as shown in Table S1. Compared with the d_S results obtained by TEM, the average d_H measured by Zetasizer showed almost 2 times.

In this study, the synthesized AgNPs have been confirmed with some functional groups stabilized on the surface, which provides extra shells to the AgNPs, leading to a larger d_H through DLS method than the d_S through TEM³⁷ as shown in Fig. S2, and similar results have been reported in previous studies.^{34, 59, 60}

Table S1 Diameter of synthesised AgNPs with different $[AgNO_3]$ measured by Zetasizer

$[AgNO_3]$ (g/mL)	AgNPs diameter (nm)	SD (nm)
0.10	16.9	5.3
0.20	18.4	4.8
0.30	20.1	4.4
0.50	24.2	6.5

Fig. S3 Schematic diagram of (a) electric double layer and (b) d_H measured by DLS and d_s measured by TEM for the AgNPs.

Comment # 9

Improve the visibility of all Figures.

We appreciate your suggestion, by which we could present the results in a clearer way. We have provided figures with high resolution in the word file, there might be some problems with the resolution when converting to PDF format. All the figures provided

in the manuscript are vector diagrams.

Comment # 10

Almost all characterization results are superficially explained. It is a need to explain in more detail with strong evidence.

Our interpretations of the characterisation results may not be sufficient in view to the theory and double check process using other characterisation methods. But, we think that we modified some unclear discussions better and our interpretations explain well the phenomena occurring during the preparation processes and properties of the AgNPs products. If you point out any insufficient and uncertain explanations additionally, we would be pleased to correspond to them.